# Role of Extracellular Vesicles in TSC Renal Cystogenesis

**DOI:** 10.3390/ijms26073154

**Published:** 2025-03-28

**Authors:** Kamyar Zahedi, Mackenzie Morgan, Brenda Prieto, Marybeth Brooks, Tamara A. Howard, Sharon Barone, John J. Bissler, Christos Argyropoulos, Manoocher Soleimani

**Affiliations:** 1Division of Nephrology, Department of Internal Medicine, University of New Mexico Health Sciences Center, Albuquerque, NM 87131, USA; memackenzie@unm.edu (M.M.); brendap1700@gmail.com (B.P.); marbrooks@salud.unm.edu (M.B.); thoward@salud.unm.edu (T.A.H.); sbarone@salud.unm.edu (S.B.); cargyropoulos@salud.unm.edu (C.A.); 2Research Services, Raymond J. Murphy Veterans Health Care Center, Albuquerque, NM 87108, USA; 3University of Tennessee Health Sciences Center, Le Bonheur Children’s Hospital, Memphis, TN 38163, USA; jbissler@uthsc.edu; 4The Department of Pediatrics, St. Jude Children’s Hospital, Memphis, TN 38105, USA

**Keywords:** extracellular vesicles, tuberous sclerosis complex, kidney, collecting duct, A-intercalated cells, principal cells, cystogenesis

## Abstract

Tuberous sclerosis complex (TSC) is caused by mutations in *TSC1* or *TSC2* genes and affects multiple organs. TSC proteins control cell growth by regulating the activity of the mechanistic target of rapamycin complex 1. Extracellular vesicles (EVs) are membrane-bound particles produced by cells that mediate cellular communication, function, and growth. Although extensive studies regarding the genetic basis of TSC exist, the exact mechanism contributing to its pathogenesis remains unresolved. It has been proposed that EVs generated by renal cyst epithelia of mice and cells with *Tsc* gene mutations contain factors that alter the function and proliferation of TSC-sufficient cells. To test this, EVs from the kidneys and kidney explants of wildtype and *Tsc1*KO mice were isolated and characterized by Western blotting, transmission electron microscopy, dynamic light scattering, and fluorescent nanoparticle tracking. Our results show an enrichment in EV-associated markers and particle sizes of similar ranges. RNA-seq and proteomic analyses identified EV shuttle factors. EV RNA and protein shuttle factors showed significant differences. Furthermore, EVs isolated from *Tsc1*KO mice inhibited the proliferation of M-1 cells. Understanding the role of EVs in cell proliferation and cystogenesis in TSC may lead to the development of new approaches for the treatment of this disease.

## 1. Introduction

Tuberous sclerosis complex (TSC) is an autosomal dominant disease that is caused by mutations in *TSC1* or *TSC2* genes [1,2,3,4]. These mutations affect hamartin (TSC1) and tuberin (TSC2) proteins [2,5], which along with TBC1D7 form a trimolecular complex that regulates the activity of the mechanistic target of rapamycin complex 1 (mTORC1) and controls the cellular response to the environment, thereby affecting cell growth and proliferation [3,6,7]. TSC affects multiple organs, including the brain, lungs, heart, skin, and kidneys [8]. Approximately 85% of individuals with TSC have renal lesions (e.g., renal cysts, cystadenomas, and angiomyolipomata), and nearly half of these individuals present with multiple renal cysts [9,10]. The renal lesions in TSC disrupt the renal parenchyma and may eventually lead to renal failure [9,11,12]. Examination of cysts in individuals with TSC and in mouse models of TSC indicate that these lesions are primarily composed of A-intercalated cells (A-ICs) that express both TSC1 and TSC2, suggesting that the mechanism of renal cystogenesis is not completely congruent with the genetic explanation of TSC [13,14,15,16]. This becomes more apparent considering that the renal cyst epithelia in mice with cell-specific knockout of *Tsc1* or *Tsc2* genes in principal cells (PCs) or pericytes are almost entirely composed of A-ICs that express both TSC1 and TSC2 [13,14].

The loss of heterozygosity (LOH) has been the accepted mechanism behind the pathology of TSC-associated hamartomas [17,18]. While LOH is confirmed in TSC-associated hamartomas [19,20,21,22], it is not always observed in other TSC lesions, such as renal cysts [13,14,23,24]. Other studies have noted that the LOH of *TSC1* or *TSC2* genes does not explain the pathologies of all TSC lesions, pointing to the need for the examination of other pathogenic mechanisms. A potential mechanism that can explain this discrepancy is that EVs generated by the cyst epithelia play a role in the mediation of cellular changes leading to dysregulated growth and expansion of TSC-sufficient cells [14,25,26]. Defining the LOH-independent mechanism in TSC etiology and the role of EVs in its mediation will expand our understanding of the non-canonical mechanisms of TSC pathogenesis.

EVs are membrane-bound particles that are produced by cells and are important regulators of cell growth, function, and communication [27,28]. The role of EVs in the functional adaptation of the cortical collecting duct (CCD) epithelium to a variety of stimuli has been demonstrated both in vitro and in vivo [29]. In TSC, the role of EVs in imparting the cystic phenotype to genetically normal cells has been documented [25,26]. Recent studies also indicate that EVs derived from renal inner medullary collecting duct cell lines with mutations in *Tsc2* genes are involved in the regulation of cell growth and that EVs play a potentially important in renal cystogenesis, whereas EVs derived from wildtype (WT) or *Tsc1*-mutant cells did not show such activity [30].

The studies presented here characterize the EVs derived from the kidneys and kidney explants of WT and principal cell-specific *Tsc1*-knockout mice (*Tsc1*cKO/Aqp2cre; *Tsc1*KO) mice. They compare the contents of shuttle factors (e.g., lncRNAs, miRNAs, snRNAs, and proteins) and the biological effects of EVs derived from the kidneys and kidney explants of WT and *Tsc1*KO mice.

## 2. Results

**Isolation and characterization of EVs.** Previous studies indicate that mutations in TSC1 or TSC2 not only lead to disease phenotypes of different severity but that they also differentially affect the production of EVs [30]. While the contents and the roles of EVs from mice and cell lines with ablated *Tsc1* or -*2* genes have been examined [26,30,31], the EVs derived from the kidneys of *Tsc1*KO mice have not been fully studied. In the work presented here, EVs isolated from pulverized kidneys or kidney explants from WT and *Tsc1*KO mice were characterized. Western blot analyses revealed that all preparations were enriched for the EV-associated markers CD63 and RAB27A (Figure 1A). We compared kidney sections from *Tsc1*KO and WT mice for the presence of CD63. Our results demonstrate the presence of intense subapical CD63 staining and its co-localization with apical H+-ATPASE in the cyst epithelia of *Tsc1*KO but not WT mice (Figure 1B).

The EVs isolated from kidneys and kidney explants of WT and *Tsc1*KO mice were subjected to TEM, DLS, and fluorescence nanoparticle tracking (fNTA) studies to confirm their purity and physical characteristics. The TEM analysis of samples isolated from kidneys and kidney explants revealed the presence of EVs with a cup-shaped outline that ranged in size from approximately 50 to 250 nm (Figure 2A,B). The sizes of the EVs in all the samples tested seemed to be comparable.

The DLS analysis of EVs obtained from the kidneys of WT and *Tsc1*KO mice (Figure 2C) revealed them to be of similar size (136 + 60.8 nm and 134.4 + 55 nm) and number (1.6 × 1011 and 1.9 × 1011). The examination of kidney-explant-derived EVs (Figure 2C) also demonstrated that the EVs from WT or *Tsc1*KO mice were of similar size (158.6 + 49.8 nm and 159.9 + 50.5) and number (7.1 × 1010 and 6.4 × 1010).

The fNTA results comparing the kidney- and kidney-explant-derived EVs from *Tsc1*KO and WT mice further confirmed that, regardless of the strain of mouse (*Tsc1*KO vs. WT) or the source (kidney or kidney explant), there were no significant differences in the size or concentration of EVs (Figure 2C). The isolated EV size in all samples was between 50 and 300 nm, with the majority of EVs falling in the 110–140 nm range. These results further demonstrated that the EV preparations had a purity that exceeded 65%.

**Comparison of the shuttle transcriptomes of EVs isolated from Tsc1KO vs. WT kidneys.** EVs are important in the regulation of cell growth and function [27,28]. In TSC, EVs are mediators of phenotype spreading, where they induce an abnormal phenotype in genotypically normal cells [26,30]. It has also been shown that the EVs isolated from parental and *Tsc2*-knockout cell lines have different biological functions and carry distinct shuttle factors [32,33]. Comparison of TSC2- and TSC1-deficient cells also reveals these cells to produce EVs that are quantitatively and functionally (e.g., with respect to mediation of a proliferative response) different. To develop a better understanding of the alterations in the shuttle factors carried by EVs isolated from WT and *Tsc1*KO kidneys, we analyzed and compared their transcriptomes. Comparison of the shuttle RNAs in the EVs of *Tsc1*KO and WT mice revealed a total of 140 differentially represented transcripts, which included both coding and non-coding RNAs (Figure 3A and Appendix A). Specifically, the 140 differentially expressed transcripts included 19 miRNAs, 24 lncRNAs, 1 vtRNA, 107 mRNAs, and 8 pseudogenes. All the differentially expressed lncRNAs except three were unannotated and without an ascribed function.

Enrichment analysis revealed multiple biological pathways that are enriched in the shuttle transcriptome of *Tsc1*KO mouse EVs, with the increases of greatest significance in the mTOR signaling, phospholipase D signaling, and glycerophospholipid metabolism pathways (Figure 3B). Within the enriched pathways, the mRNA coding for *Slc25a26*, a mitochondrial transport protein responsible for S-adenosylmethionine (SAM)/S-adenosylhomocysteine (SAH) exchange, and *Lpin2*, a phosphatidic phosphatase, were significantly present in the EVs of *Tsc1*KO mice.

**Comparison of the shuttle proteomes of EVs isolated from Tsc1KO vs. WT kidneys.** The shuttle proteomes of WT vs. *Tsc1*KO kidney-explant- and kidney-derived EVs were analyzed and compared (Appendix A). Examination of kidney-explant-derived EV proteomes of *Tsc1*KO and WT mice was performed using the FragPipe-Analyst online tool (http://fragpipe-analyst.nesvilab.org/; accessed on 25 August 2024). These analyses revealed the presence of 98 proteins that differed between the two sets of samples. Principal component analysis (PCA) revealed clear distinctions between the proteomes of *Tsc1*KO and WT kidney-explant-derived EVs (Figure 4A). The examination of the heat map demonstrated distinct groupings in the protein contents of EVs when WT and *Tsc1*KO samples were compared (Figure 4B). A similar distinction between the proteomes of the two EV populations was apparent in the volcano plot of the results (Figure 4C).

Comparison of EV proteomes from kidneys of WT and *Tsc1*KO mice showed the presence of 124 proteins whose expression was significantly different. The PCA results indicated the presence of distinct clusters when the kidney EV proteomes from *Tsc1*KO and WT mice were compared (Figure 5A). Heat map and volcano plot results also demonstrated distinct groupings in the protein contents of EVs when WT and *Tsc1*KO samples were compared (Figure 5B,C).

Proteins that were enriched in the EV preparations derived from *Tsc1*KO kidneys and kidney explants were associated with proton extrusion, vesicular transport, regulation of cytoskeleton assembly, and cell proliferation (Appendix A). Enrichment analysis of the differentially represented proteins in *Tsc1*KO-derived EVs did not identify any significantly (FDR < 0.05) enriched pathways.

**Effect of EVs isolated from the kidneys of Tsc1KO and WT mice on cell growth.** The effect of EVs derived from the kidneys of *Tsc1*KO on cell growth has not been studied. While the effect of EVs derived from *Tsc2*KO mice and *Tsc2*-deficient cell lines on cell growth has been well defined [30,31,32,33], the same cannot be said about EVs derived from the kidneys of *Tsc1*KO mice. Therefore, we determined the effect of EVs derived from the kidneys and kidney explants of WT and *Tsc1*KO mice on the growth of M-1 mouse CCD cell lines. To exclude the effect of EVs present in serum, the EV preparations were made in the absence of any extraneous EV sources (e.g., serum), and the M-1 cells used in these studies were adapted for growth in serum-free medium. The effect of EV preparations from WT and *Tsc1*KO samples on subconfluent (70% confluent) monolayers of M-1 cells was examined at 24 and 48 h. Our results also revealed that EVs from kidneys or kidney explants derived from Tsc1KO mice significantly reduced the growth of M-1 cells compared to those from WT mice (Figure 6).

## 3. Discussion

TSC kidney lesions, including cysts, can lead to severe damage to the parenchyma and eventually lead to renal failure [9,11,12]. Studies show that in TSC cystic epithelial cells express both TSC1 and TSC2 proteins [13,14] and therefore do not adhere to the LOH genetic mechanism ascribed to TSC-associated angiomyolipmas [8,11,18,34]. For example, deletion of the *Tsc2* gene in mouse principal cells leads to the development of cysts that are almost entirely composed of A-IC cells that still express TSC2 [14]. Similarly, in principal cell-specific *Tsc1*KO mice the cysts are primarily composed of A-IC cells that express both TSC1 and TSC2 [13]. These results follow what is observed in individuals with TSC renal cystic disease, where A-IC cells are by far the predominant cell type found in the cystic epithelium, and there is no loss of heterozygosity [15]. In fact, individuals with the polycystic variety of TSC renal disease continue to express both tuberin and hamartin in their cyst epithelia. Remarkably, individuals diagnosed with PKD1-associated ADPKD also continue to express polycystin-1 in the renal cystic epithelium [35,36]. These observations suggest that phenotypic changes that are independent of the genetic makeup of the cells drive the proliferative response behind cystogenesis in TSC and possibly PKD. We previously proposed that such changes may be mediated through EVs [14]. Street et.al. have demonstrated that cellular functions in CCD can be modified by EVs isolated from urine [29]. Additional studies have expanded on the role of EVs in TSC-associated pathologies in the brain and kidneys [26,30,33]. These studies identified the role of EVs obtained from both *Tsc2*-mutant cell lines and *Tsc2*KO mice in the gain of a TSC disease phenotype by genotypically normal cells [26,30,31,33]. However, the potential role of kidney-derived EVs in the development of renal lesions in animals with *Tsc1* mutation has not been scrutinized.

The studies described here defined the physical characteristics, structure, shuttle contents, and biological activity of EVs derived from kidneys and kidney explants of WT and *Tsc1*KO mice. The *Tsc1*KO mice were developed through principal cell-specific ablation of the *Tsc1* gene using a constitutively expressed Cre recombinase transgene that is under the control of a modified aquaporin 2 gene promoter [13]. These mice developed renal cysts that were primarily composed of A-IC cells that express both TSC1 and TSC2 [13]. A similar preponderance of A-IC cells in the cystic epithelium was observed in mice where the *Tsc1* or *Tsc2* genes were knocked out in either pericytes or principal cells, respectively, as well as in heterozygotic *Tsc2*KO mice [14,37]. The above results, in addition to what has been described in the renal biopsies of TSC patients [15], indicate that A-IC cells are important in the development of renal cysts.

Our results demonstrate that the EVs isolated from the kidneys and kidney explants of WT and *Tsc1*KO mice displayed similar physical properties as determined by TEM, DLS, and fNTA analyses (Figure 2). Furthermore, the EV numbers in preparations from the kidneys and kidney explants of WT and *Tsc1*KO mice were not significantly different (Figure 2). All EV preparations showed enrichment for CD63 and RAB27A markers (Figure 1A). Whereas the quantitative and physical characterization of the EV preparations from WT and *Tsc1*KO kidneys did not reveal any significant differences, our results clearly indicated that their shuttle contents and biological activities are quite disparate.

Comparison of CD63 and H+-ATPASE localization in the cyst epithelia of *Tsc1*KO mice revealed that CD63 levels were significantly elevated in the subapical region of H^+^-ATPASE-expressing cells; in comparison, the expression of CD63 was significantly lower in the normal epithelia of WT kidneys (Figure 1B). Whether the presence of CD63 in the subapical region of cystic epithelial cells indicates enhanced generation and/or increased retention of EVs because of mutations in the *Tsc1* gene is not clear. The subapical localization of CD63 in A-IC cells that lack primary cilia may be functionally relevant given the role that the primary cilia play in EV extrusion [38], especially because the A-IC cells that make up the cyst epithelium lack primary cilia [39]. The mechanism of EV extrusion by A-IC cells that are independent of the primary cilia remains to be addressed.

Previous studies have shown that the extrusion of EVs by WT cells and cells with disrupted *Tsc1* is lower than that of their counterparts with a *Tsc2* mutation [31]. The quantitative difference between the EVs extruded by *Tsc1-* vs. *Tsc2*-KO cells may be an indication of how these mutations affect the production and elaboration of EVs and how such differences affect the disease phenotype associated with *TSC1* vs. *TSC2* mutations [31]. It is known that *TSC1* mutations cause a less severe disease phenotype in humans compared to *TSC2* mutations [2,40,41]. Knockout of *Tsc2* vs. *Tsc1* in mice reveals that the *Tsc2*KO mice develop a more severe cystic phenotype than *Tsc1*KO mice [13,14,16,42]. The mechanism responsible for the differences in the severity of TSC1- and TSC2-associated disease has yet to be elucidated. These variations in severity may be due to the following factors: (1) The absence of functional TSC2 (a GTPase activating factor that is responsible for the effector function of TSC): Mutations in TSC2 can then lead to RHEB and mTORC1 hyperactivation and unregulated cell growth. (2) The indirect role of TSC1 in regulating the stability of the TS protein complex: In TSC1 mutations, TSC2 activity may persist and regulate the activity of RHEB and mTORC1, albeit at a reduced level. (3) Both mutations can affect the production and shuttle content of the EVs, thereby exerting variable impacts on the surrounding epithelia. Studies investigating these factors will focus on the quantitative and qualitative comparison of EVs produced by *Tsc1*KO vs. *Tsc2*KO mice, as well as the assessment of differences in their shuttle factors and functional properties. Results from these studies will be important in defining the evolution of EV-associated factors and their contribution to the severity of TSC pathology in the disease caused by *Tsc1* vs. *Tsc2* gene mutations.

Characterization of the shuttle transcriptomes of *Tsc1*KO- compared to WT-derived EVs revealed the presence of 140 differentially expressed transcripts (Appendix A). These included 24 lncRNAs, regulatory molecules the majority of which were unannotated [43,44]; however, 3 lncRNAs, *Miat*, EZH2-associated lncRNA, and *Rncr4*, were identified as having defined biological functions. Therefore, reduced levels of *Miat* and silencing of its pro-proliferative activity may be protective against renal cystogenesis, which is driven by enhanced cell proliferation [45,46,47,48,49,50,51,52,53,54,55]. A decrease in *Miat* lncRNA levels in the kidney-derived EVs of *Tsc1*KO mice may explain the anti-proliferative activity, as well as the diminished severity of renal cystic disease in *Tsc1*KO mice. In support of this, *Miat* lncRNA was shown to enhance proliferation and invasiveness in a variety of cancer cells [45,46,47,48,49,50]. *Miat* functions as a sponge for miR184, miR214, and miR613 in order to enhance cell proliferation, but when its levels are decreased it leads to a reduction in cell growth and invasion [46,50,51]. The EZH2-associated lncRNAs are known to interact with histone methyl transferase EZH2 to promote cell proliferation, tumor progression, and cancer metastasis [52,53,54]. Furthermore, EZH2 inhibition exerts an antineoplastic effect against renal cell carcinoma through inactivation of large tumor suppressor 1 (LATS1) [55]. Thus, a decrease in the levels of EZH2-associated lncRNAs may interfere with the activity of this methyltransferase and act as an additional factor that downregulates cell proliferation and, by extension, cyst growth. *Rncr4* facilitates the processing of pri-miR-183/96/182 [56]. This miRNA complex is a known regulator of the immune response and a potential factor contributing to renal cell carcinoma development [57]. *Miat*, EZH2-associated lncRNA, and *Rncr4* may contribute to the reduction in the severity of renal cystic disease in *Tsc1*KO mice; however, the functions of these lncRNAs need to be further examined to define their role in the regulation of cystogenesis in *Tsc1*KO mice and TSC renal cystic disease in general. We also identified 107 differentially expressed mRNAs, which included transporters (Slc25a6), enzymes (Lpin2 and Bace2), and actin cytoskeleton components (Capzb). Although SLC25A6 and LIPIN2, through their metabolic role, regulate mTOR activity [58,59], it is hard to speculate on the role of β-secretase-2 (BACE2) or the actin-binding protein (CAPZb). Other non-coding RNAs identified included 19 miRNAs, 8 pseudogenes, and 1 vault RNA (vt-RNA). While the function of pseudogenes is not clear and the changes in miRNA levels were not significant, vt-RNAs, which are important regulators of cell metabolism and proliferation [60,61,62], were significantly more abundant in the EVs derived from Tsc1KO mice. The elevated presence of vt-RNA and elevated VAULT1/MVP protein levels in the EV proteomes of *Tsc1*KO mice suggests an increase in the levels of VAULT complexes and their associated functions, including the regulation of cell proliferation [63]. Thus, the MVP ribonucleoprotein complex was shown to inhibit cancer cell proliferation and survival via the inhibition of STAT3 signaling and enhancement of HIF1α [64]. The enrichment analysis of the upregulated transcripts in *Tsc1*KO-derived EVs indicated the presence of pathways involved in the regulation of mTORC1 signaling, cell proliferation, cell adhesion and structure, as well as metabolism (Figure 3B). The enriched pathways identified in these studies seem to correlate closely with changes in cell growth, as well as the transcriptome and metabolome changes in the kidneys of *Tsc1*KO mice [65].

We also examined the shuttle proteomes of EVs derived from kidneys or kidney explants of WT and *Tsc1*KO mice. Our results indicated that the shuttle proteomes of the WT and *Tsc1*KO kidney- and kidney-explant-derived EVs were significantly different (Figs. 4 and 5). The shuttle proteomes of *Tsc1*KO-derived EVs exhibited elevated levels of H+-ATPase components, such as ATP6V0d2, ATP6V0G3, and ATP6V1C2, as well as several proteins that belong to the S100 family of proteins, including S100-A1, which is a growth-inhibitory molecule that interacts with p53 and MDM2 to decrease cell proliferation [66,67]. S100-A1 is also a diagnostic marker of intercalated cell-derived chromophobe renal cell carcinomas [68,69]. The presence of H+-ATPase components and S100-A1, in addition to the absence of principal cell-associated proteins, suggests that the EVs in *Tsc1*KO are primarily derived from A-IC cells, unlike the EVs in *Tsc2*KO cells and *Tsc2*-deficient mice, which are derived from principal cells and contain high levels of markers associated with primary cilia [33]. Additional examination of the proteomes of the EVs indicated the presence of molecules with a GTPase-activating property (e.g., IQGAP1 and 2), which may function in the regulation and activity of small GTPases such as RHEB, an activator of mTORC1, which is regulated by TSC [70].

Additional studies examining the effect of EVs derived from kidney explants or whole kidneys of WT and *Tsc1*KO mice on cell growth showed that the EVs from *Tsc1*KO mice slowed down the expansion of cultured M-1 cells. Our results complement the finding of Kumar et. al. that EVs from *Tsc1*-deficient cells reduced the activation of mTORC1 and enhanced the expression of miR-212-3p and miR-99a-5p [30], which have been shown to reduce the growth of a variety of cancers [71,72]. Studies of EVs isolated from the kidneys of *Tsc1*KO mice vs. those derived from cultured cells, while informative, cannot be directly compared. The EVs tested by Kumar et. al. were derived from homozygotic *Tsc1*- or *Tsc2*-knockout cells, whereas the EVs in the present study were isolated from cells that expressed both TSC1 and TSC2. Future studies comparing the EVs derived from *Tsc1*KO vs. *Tsc2*KO mice will be conducted to determine the differences in their physical characteristics, shuttle contents, and biological activities.

The genetic mechanism of TSC is based on initial mutation in either of the *TSC1* or *TSC2* loci followed by a second hit which inactivates the second locus of the affected gene [11]. Although this mechanism applies to the development of AML in the kidneys [11,18,34], it does not apply to TSC renal cysts or brain tubers [13,14,15,16,20,73,74,75]. In fact, examination of renal cysts in mouse models of TSC indicates that they are composed of cells that express both TSC1 and TSC2 [13,14]. A similar genetic heterogeneity is also observed in other TSC lesions, such as brain tubers [20,73,74,75]. This anomalous behavior of genetically normal cells is proposed to be the result of phenotypic spreading that is mediated by EVs produced by TSC-mutant cells [14,26,30]. It has also been demonstrated that EVs derived from *Tsc2*-mutant cells carry shuttle factors (proteins and miRNA) that regulate the proliferative activity of TSC-sufficient cell lines [30,31]. Our initial studies demonstrated that the cystic epithelia in mice with principal cell knockout of *Tsc1* is composed of A-ICs that express both TSC1 and TSC2 [13,14]. The principal cell-specific *Tsc1*-knockout (*Tsc1*KO) mouse model was used to determine the EV shuttle factors and the role of EVs isolated from the kidneys and kidney explants of *Tsc1*KO mice in renal cystic disease. While the number and physical characteristics of EVs isolated from the WT and *Tsc1*KO mice were similar in our studies, this was shown not to be the case for the *Tsc2*KO mice, where the interstitial EV content was greater and the EV size was larger than that of WT mice [30,31]. Our results show (1) the potential retention or hindered extrusion of EVs in the kidneys of *Tsc1*KO mice, (2) the reduced presence of shuttle factors with pro-proliferative activity in the proteomes and transcriptomes of EVs isolated from *Tsc1*KO mouse kidneys, and (3) the anti-proliferative activity of these isolated EVs. Together, these findings suggest that *Tsc1*KO-derived EVs harbor growth-inhibitory properties. The latter characteristics make the *Tsc1*KO-derived EVs functionally distinct from those that are derived from *Tsc2*-deficient cells and *Tsc2*-mutant mice [30,31]. The EVs characterized in these studies were derived from mice with advanced renal cystic disease and thus represent a specific time frame in the disease process that is characterized by a preponderance of A-ICs in the epithelia of established cysts. It is likely that the qualitative and functional characteristics of the EVs evolve as the disease progresses and that the affected epithelia change and adapt to the disease process. Furthermore, comparison of EVs derived from *Tsc1*KO and *Tsc2*KO mice should lead to a better understanding of the differences in their characteristics, contents, and biological functions. These studies will potentially address the differences in the severity of the disease caused by mutations in *TSC1* vs. *TSC2* mutations.

The current study characterized the EVs derived from the kidneys of *Tsc1*KO mice. Examination of EV-associated markers suggested that there is an accumulation of CD63 in the apical aspect of the cyst epithelium. Whether this is due to the sequestration of EVs or their increased generation is not clear. The anti-proliferative effect of EVs identified in this study may be an adaptive response and a function of the advanced stage of renal cystic disease. Future studies comparing the evolution of EVs and the differences in *Tsc1*KO- vs. *Tsc2*KO-derived EVs will address the role of EV-mediated cellular changes in the progression of TSC lesions. In addition, identifying and deciphering the role of specific shuttle factors associated with EVs from different time points or different strains should lead to the development of targeted therapies for the treatment of TSC renal disease.

## 4. Materials and Methods

### 4.1. Animals

*Tsc1*KO mice have the *Tsc1* gene inactivated in principal cells. In this study, such mice were derived by crossbreeding Tsc1tm1Djk/J (*Tsc1*cKO) and B6.Cg-Tg(Aqp2-cre)Dek/J (*Aqp2*Cre) mice. The PCR genotyping protocol of animals was performed as previously described [13,14]. For these studies, *Tsc1*KO mice and their WT littermates were euthanized at 42–45 days of age and their kidneys were harvested. For isolation of kidney EVs, the organs were pulverized, passed through a 100 µm strainer, Dounce-homogenized, and then processed for EV purification. For isolation of EVs, we used a previously established protocol [33,76]. Briefly, harvested kidneys were decapsulated and cut into approximately 2 mm^3^ sections, washed in hanks balanced salt solution, resuspended in serum-free medium (Corning; Corning, NY, USA), and incubated in 35 mm tissue culture dishes at 37 °C in 5% CO_2_. The content of each dish was collected and centrifuged at 1500× *g* for 5 min to remove the particulate matter and then processed for EV extraction.

### 4.2. Preparation of EVs

The EV extraction protocol was a modification of the method described in [33,76]. Briefly, crude preparations were subjected to centrifugation at 3000× *g* for 30 min, 5000× *g* for 30 min, and 10,000× *g* for 30 min at 4 °C. The preparations were then passed through a 0.8 µm syringe-top filter (Merk Millipore; Carrigtwohill, County Cork, Ireland). The filtered homogenates were concentrated using Amicon (Ultracel 100K) centrifugal filters (Amicon; Carrigtwohill, County Cork, Ireland). The concentrated samples were brought up to a volume of 200 µL with PBS. The isolation of EVs (150 µL of each preparation) was performed by size fractionation using IZON qEV35nm single-use columns (IZON, Bellaire, TX, USA).

### 4.3. Western Blot Analyses

Isolated EV preparations were denatured in Laemmli sample buffer (Bio-Rad Laboratories Inc.; Hercules, CA, USA) containing dithiothreitol, size-fractionated by SDS/PAGE, and transferred to nitrocellulose filters. Western blot analyses were performed using antibodies that recognize mouse CD63 (Abcam; Waltham, MA, USA) and mouse RAB27A (Abcam; Waltham, MA, USA), protein markers of EVs.

### 4.4. Immunofluorescence Microscopy

Kidneys were fixed in 4% paraformaldehyde, preserved in 70% ethanol, and paraffin-embedded. Paraffin-embedded sections were cut into 5 um sections and underwent antigen retrieval, employing a 2100 Retriever (Electron Microscopic Sciences; Hatfield, PA, USA). Sections were blocked in PBS containing normal horse serum, 0.2% powdered skim milk, and 0.3% Triton X-100 for at least 60 min at room temperature before incubation with anti-CD63 and anti-ATP6V1B1/B2 (Santa Cruz; Dallas, TX, USA) at 4 °C for 24 h. Slides were washed in PBS for 3 × 10 min and incubated in Alexafluor secondary antibodies (Invitrogen; Eugene, OR, USA) at room temperature for 2 h. After drying, a coverslip was applied using VectaShield Hard-Set (Vector Labs; Newark, CA, USA) as the mounting medium. Images were obtained with a Zeiss LSM800 microscope utilizing Zen software (Version 3.4.91.000, Carl Zeiss, Oberkochen, Germany).

### 4.5. Transmission Electron Microscopy (TEM)

Isolated EV preparations were negatively stained using uranyl acetate, and their size and morphology were examined by TEM. Briefly, 10 µL of freshly isolated EV suspension was incubated on a freshly glow-discharged carbon-coated grid at room temperature. The samples were washed with ultrapure water to remove buffer salts, then negatively stained using 2% uranyl acetate and air-dried. The TEM analyses were performed at the UNM-HSC Electron Microscopy facility using a Hitachi HT7700 instrument (Chiyoda, Japan) equipped with an AMT XR16M 16-megapixel digital camera (AMT Imaging Direct; Woburn, MA, USA).

### 4.6. DLS and fNTA Analysis of EV Preparations

EV preparations were analyzed using the services provided by Alpha Nanotech LLC (Durham, NC, USA). These studies consist of the labeling of intact exosome membranes with a fluorescent dye (Cell Mask Deep Red, CMDR; Thermo Fisher Scientific, Carlsbad, CA, USA) and then performing the analysis in scatter and fluorescent modes. The analyses were performed on a Zetaview Quatt LASER instrument (Particle Metrix; Mebane, NC, USA).

### 4.7. Examination of EV Shuttle Transcriptomes

The EV transcriptomes were assessed through a novel RNA sequencing protocol adapted to permit simultaneous detection of coding and non-coding transcripts, utilizing an Oxford Nanopore Technologies (ONT) long-read sequencer [77]. Transcripts were mapped to the mouse transcriptome for differential expression and pathway enrichment analysis.

### 4.8. Examination of EV Shuttle Proteomes

The shuttle proteomes of EVs were analyzed by the UNM Proteomic Core Facility. Analysis of the kidney- and kidney-explant-derived EV proteomes of Tsc1KO and WT mice was accomplished using the FragPipe online tool (http://fragpipe-analyst.nesvilab.org/; accessed on 25 August 2024).

### 4.9. Cell Proliferation Assay

The mouse CCD-derived M-1 cell line (ATCC; Manassas, VA, USA) was used in these studies. Due to the presence of extracellular vesicles in fetal bovine serum, the M-1 cells were adapted to grow in serum-free medium (M-1 SFM). The M-1 SFM cells were seeded at 5 × 10^5^ cells/well in 12-well tissue culture dishes and incubated in a humidified cell culture incubator (37 °C, 5% CO_2_). Upon reaching 70% confluency, the cells were left untreated or treated with 5 µg/well of EVs. For cell enumeration, cells were washed with PBS, stained with crystal violet, washed with PBS, and solubilized in 1%SDS. The released stain was quantified by measuring the absorbance of the SDS solution at 590 nm.

### 4.10. Statistical Analysis

The statistical differences between mean values ± SDs of multiple samples were determined using a two-tailed unpaired Student’s *t*-test. A *p*-value of less than 0.05 was considered statistically significant.

## Figures and Tables

**Figure 1 ijms-26-03154-f001:**
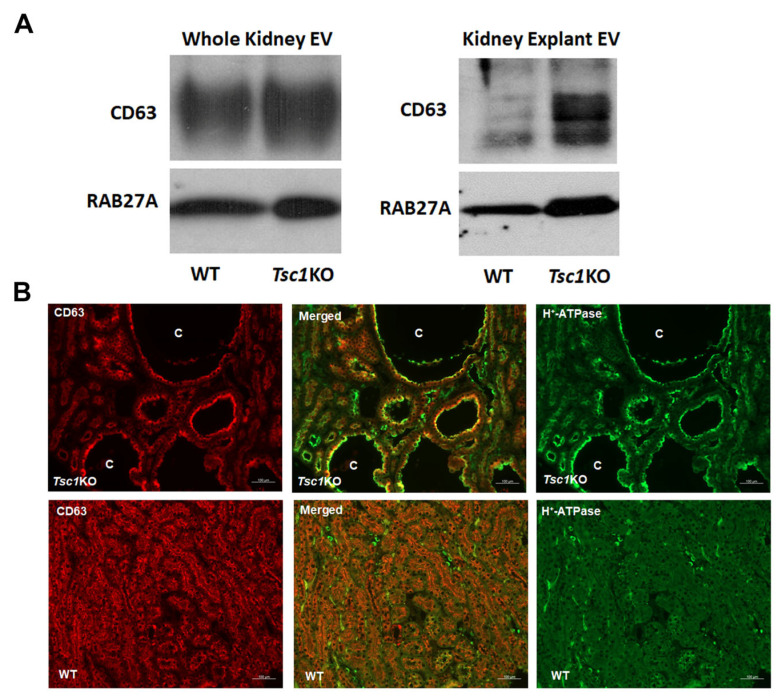
Western blot and immunofluorescence microscopic analyses of specific EV markers. The expression of CD63 and RAB27A was examined by Western blot analysis, and the localization of CD63 was determined by immunofluorescence microscopy in the EVs and kidneys of WT and *Tsc1*KO mice. (**A**) The EVs isolated from kidneys and kidney explants of WT and *Tsc1*KO mice were subjected to Western blot analysis for the presence of CD63 and RAB27A. The results show that EV preparations were enriched for both markers. (**B**) Kidneys of WT and *Tsc1*KO mice were examined for the expression and localization of CD63 and H+-ATPASE (ATP6 V1B1/2). While CD63 expression was minimal in the CCD of WT mice, its expression was upregulated in the cystic epithelia of *Tsc1*KO animals. Furthermore, CD63 was localized to the subapical region of cystic epithelial cells that also expressed H+-ATPASE on their apical membrane (A-ICs). “C” represents cysts. Scale bars represent 100 μm.

**Figure 2 ijms-26-03154-f002:**
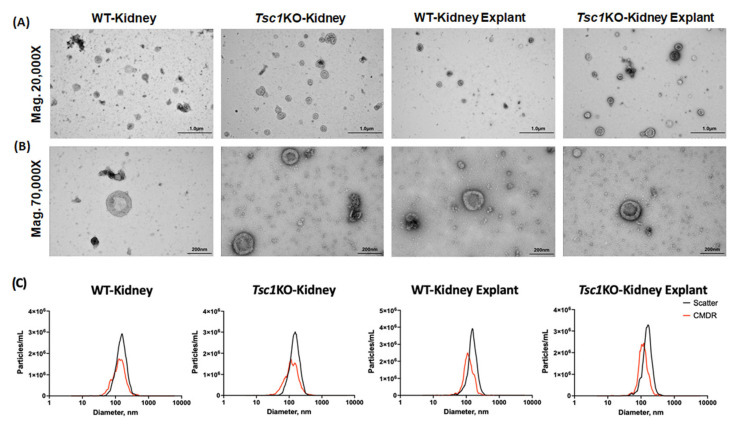
Physical analysis of EV preparations. (**A**) Representative TEM images (20,000× magnification) of isolated EVs from WT and *Tsc1*KO kidneys and kidney explants. (**B**) Higher-magnification (70,000×) images of EVs from kidney and kidney-explant preparations. (**C**) Representative results of DLS (black) and fNTA (red) characterization of EV preparations. The EVs demonstrated the expected cup-shaped outline. The size of EVs isolated from all samples ranged from 50 to 250 nm. The EVs in the 110–150 nm range were the most prevalent.

**Figure 3 ijms-26-03154-f003:**
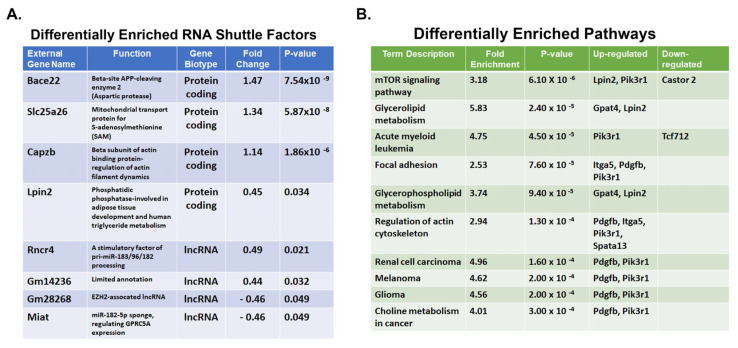
Examination of EV transcriptomes. The shuttle transcriptomes of EVs isolated from kidney and kidney explants of WT and *Tsc1*KO mice were analyzed and contrasted. The transcriptome profiles were enriched in protein-coding and long non-coding (lnc) RNAs (Appendix A). (**A**) List of the most significantly upregulated RNAs in *Tsc1*KO (left) relative to WT mice. (**B**) The EV preparations from *Tsc1*KO animals are enriched for RNAs coding for molecules that are involved in mTOR regulation and lipid metabolism.

**Figure 4 ijms-26-03154-f004:**
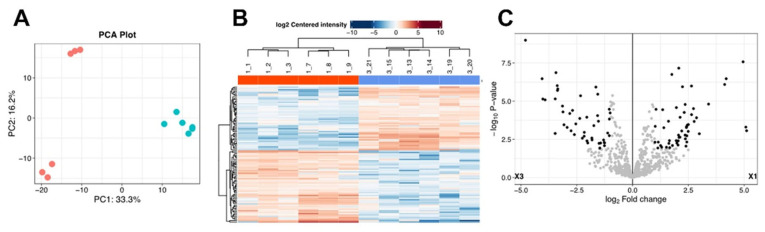
Analysis of the kidney-explant-derived EX shuttle proteomes. The proteomes of EVs isolated from kidney explants of WT and *Tsc1*KO mice were analyzed and compared (Appendix A). Comparison of shuttle proteomes was performed using the Frag-Pipe online tool (http://fragpipe-analyst.nesvilab.org/; accessed on 25 August 2024). (**A**) The PCA plot shows the clustering of EV samples prepared from the kidney explants of WT and *Tsc1*KO mice. The orange symbols represent kidney explant exosomes derived from *Tsc1*KO mice, while the blue symbols represent kidney explant exosomes derived from WT mice. (**B**) Heat map demonstrating the differences in the proteomes of EV samples from the kidney explants of WT and *Tsc1*KO mice. (**C**) Volcano plot of the shuttle proteomes of EV samples from the kidney explants of WT and *Tsc1*KO mice. The designations X1 and X3 refer to the proteome of kidney explant exosomes derived from *Tsc1*KO and WT mice, respectively.

**Figure 5 ijms-26-03154-f005:**
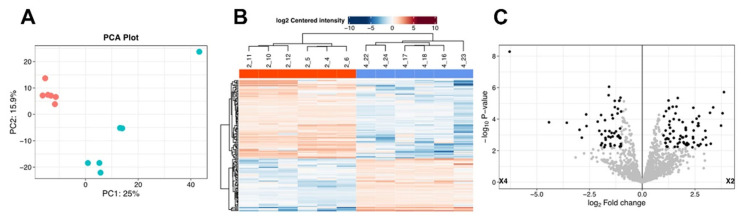
Analysis of the kidney-derived EX shuttle proteomes. The proteomes of EVs isolated from kidneys of WT and *Tsc1*KO mice were analyzed and compared (Appendix A). Comparison of shuttle proteomes was performed using the Frag-Pipe online tool (http://fragpipe-analyst.nesvilab.org/; accessed on 25 August 2024). (**A**) The PCA plot shows the clustering of EV samples derived from the kidneys of WT and *Tsc1*KO mice. The orange symbols represent kidney exosomes derived from *Tsc1*KO mice, while the blue symbols represent kidney exosomes derived from WT mice. (**B**) Heat map results demonstrating the differences in the proteomes of EV samples from the kidneys of WT and *Tsc1*KO mice. (**C**) Volcano plot of the shuttle proteomes of EV samples from the kidney explants of WT and *Tsc1*KO mice. The designations X2 and X4 refer to the proteome of kidney exosomes derived from *Tsc1*KO and WT mice, respectively.

**Figure 6 ijms-26-03154-f006:**
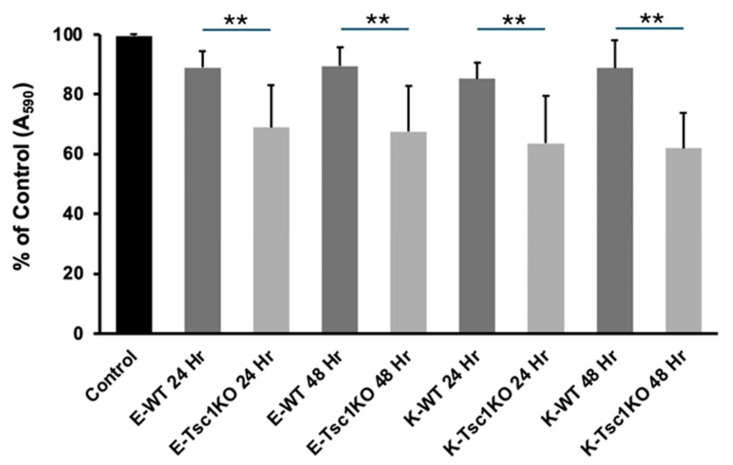
Effect of EVs on cell proliferation. The effect of kidney- and kidney-explant-derived EVs isolated from WT and *Tsc1*KO mice on the proliferation of mouse collecting duct M-1 cell lines was examined. The data presented are combined results of 2 independent experiments, with n = 4 replicas/sample/experiment. Abbreviations: E represents kidney explants, and K represents kidney-derived EVs. ** represent a *p* < 0.01 when comparing Tsc1KO derived exosomes to time-matched WT cells. “E” designates explant-derived exosomes and “K” designates kidney-derived exosomes.

## Data Availability

The datasets used and/or analyzed in the current study are included in the published article and its Appendix A. These documents are also available from the corresponding author upon request.

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
