# Peer review of "Role of Extracellular Vesicles in TSC Renal Cystogenesis"

_ijms, 2025, doi:10.3390/ijms26073154_

Round 1
Reviewer 1 Report
Comments and Suggestions for Authors
The researchers investigated how extracellular vesicles (EV) from TSC1 knockout mice differ from normal mice, finding significant differences in RNA and protein content, with EVs from TSC1KO mice showing an ability to inhibit cell proliferation. This study suggests that understanding EV's role in TSC could potentially lead to new therapeutic approaches, as TSC (caused by TSC1/TSC2 mutations) affects multiple organs through dysregulated cell growth pathways. Moreover, the result is technically sounded and worthy to be published in Int. J. Mol. Sci.
The following are some comments and suggestions that are given to improve the manuscript:
Comment 1: The paper mentioned a significant difference in exosome (EV) vectors between Tsc1KO mice and wild-type mice. Whether these differences change at different stages of disease progression can be compared by further studies to see if the role of exosomes changes as cysts develop.
Comment 2: Because Tsc1KO mice and Tsc2KO mice display different disease severity, how do differences in EV vectors contribute to the different disease phenotypes between TSC1 and TSC2 mutations?
Comment 3: It was shown that EV derived from Tsc1KO mice inhibited proliferation of M-1 cells. Could this finding be extended to other cell lines or animal models to confirm whether the anti-proliferative properties of EV are generally applicable in different types of renal cysts.
Comment 4: Current studies focus on the effect of exosomes on M-1 cell proliferation in the short term. Can Tsc1KO mouse-derived EVs be assessed for their long-term effects on cell proliferation, differentiation, or migration through long-term culture experiments, leading to a more comprehensive understanding of their regulation of kidney cell growth?

The English could be improved to more clearly express the research.
Author Response
Reviewer 1
The researchers investigated how extracellular vesicles (EV) from TSC1 knockout mice differ from normal mice, finding significant differences in RNA and protein content, with EVs from TSC1KO mice showing an ability to inhibit cell proliferation. This study suggests that understanding EV's role in TSC could potentially lead to new therapeutic approaches, as TSC (caused by TSC1/TSC2 mutations) affects multiple organs through dysregulated cell growth pathways. Moreover, the result is technically sounded and worthy to be published in Int. J. Mol. Sci.
We thank the reviewer for their constructive comments and pertinent suggestions. We believe that the quality of the manuscript has improved significantly and hope that the reviewers and the Editorial Board find the manuscript acceptable for publication.
The following are some comments and suggestions that are given to improve the manuscript:
Comment 1: The paper mentioned a significant difference in exosome (EV) vectors between Tsc1KO mice and wild-type mice. Whether these differences change at different stages of disease progression can be compared by further studies to see if the role of exosomes changes as cysts develop.
Studies described here examine EV derived from Tsc1KO mouse kidneys that showed extensive cyst burden. We agree with the reviewer that the structure, content, and function of EV may evolve as the process of cystogenesis progresses. Of course, characterizing these changes is important in the development of better understanding of the role of EV and their evolution during cyst development. Statements/explanations regarding this comment are included in lines 371-376 and 383-389.
Future studies comparing the differences in EV derived from Tsc1 vs Tsc2 knockout mouse models are important components of our ongoing work; however, they are outside of the purview of the current manuscript.
Comment 2: Because Tsc1KO mice and Tsc2KO mice display different disease severity, how do differences in EV vectors contribute to the different disease phenotypes between TSC1 and TSC2 mutations?
Although we have not directly compared the shuttle content of EV from Tsc1KO and Tsc2KO mice, studies by Kumar et. al., 2021 suggest that the EV derived from TSC1 and TSC2 deficient cells exhibit different biological activities. Specifically, the EV from TSC2 deficient cells exhibit an mTORC1 activating capacity that is not observed in the EV from TSC1 deficient cells. As such the pro-proliferative activity associated with EV from the two sources may partially explain the increased severity of the disease caused by TSC2 vs TSC1 deficiencies. Statements/explanations regarding this comment are included in lines 265-278, 340-346, 376-389.
Comment 3: It was shown that EV derived from Tsc1KO mice inhibited proliferation of M-1 cells. Could this finding be extended to other cell lines or animal models to confirm whether the anti-proliferative properties of EV are generally applicable in different types of renal cysts.
The Tsc1KO model examines the TSC renal cystic disease and targets cortical collecting duct (CCD) component of the nephron, where the significant majority of cysts originate. We have not examined the effect of Tsc1KO-derived EV in other models of TSC or non-TSC renal cystic diseases. Because of the role of CCD cells in TSC cystic disease, we concentrated on the effect of Tsc1KO- derived EV on M-1 CCD cell lines. The M-1 cells that we used in these studies were chosen for several reasons: 1) these cells were adapted for growth in serum-free medium; this is important since both fetal bovine serum and “EV depleted fetal bovine serum” contain EV that can affect the experimental results; 2) the renal cysts in Tsc1KO and other models that we have studied are derived from CCD; therefore, M-1 cells because of their CCD origin were the most logical cells to use in these studies; and 3) mIMCD3, the other collecting duct mouse cell line available, are derived from the intermedullary collecting duct, a region that is not involved in TSC renal cystogenesis. Kidney cell lines from other nephron segments will be used in future studies, but these cells need to be from areas of the kidney that are prone to cyst formation and the cells of interest need to be adapted for growth in serum-free medium. Please refer to lines 192-194.
Comment 4: Current studies focus on the effect of exosomes on M-1 cell proliferation in the short term. Can Tsc1KO mouse-derived EVs be assessed for their long-term effects on cell proliferation, differentiation, or migration through long-term culture experiments, leading to a more comprehensive understanding of their regulation of kidney cell growth?
The cell proliferation studies described here are short term growth assays. Additional studies focusing on the long-term effect of EV on M-1 cells, as well as examination of specific populations of EV are planned as a part of our future studies. We are mindful that long-term studies aimed at deciphering the effect of EV on cell proliferation, differentiation, or migration can be challenging if the cellular response to EV is a sequential process (i.e., EV from various time points lead to changes that are dependent on exposure to other initiating factors or EVs). Such a sequential response may be demanding to replicate in vitro, especially considering the reviewer’s 1st comment regarding the potential changes in EV as a function of cyst progression.
Reviewer 2 Report
Comments and Suggestions for Authors
Dear authors,
- This paper discusses how the genetic deletion of TSC1 and TSC2 affects the generation of exosomes (EVs). However, the background explanation is somewhat vague, and the importance and objectives of the research should be emphasized more. Specifically, a more detailed discussion of the distinct pathological effects of TSC1 and TSC2 would help readers better understand the significance of the study.
- The paper presents the interesting result that EVs from Tsc1KO mice exhibit anti-proliferative effects. A more detailed discussion of the mechanisms behind this, such as how specific miRNAs or proteins are involved, would make it easier to understand the role of EVs and highlight the significance of the research.
- The report mentions differences in the characteristics of EVs between Tsc2KO and Tsc1KO mice. However, a more detailed analysis of this comparison is needed. It is shown that the progression of the disease in Tsc2KO mice is more severe than in Tsc1KO mice. A more thorough explanation of the reasons for this would clarify how the different functions of EVs contribute to the severity of the disease, so this should be further explored.
- The paper currently uses a variety of techniques to collect data, including Western blot, TEM, DLS, fNTA, transcriptome analysis, and proteome analysis. However, it is not entirely clear how the data from each technique are integrated and how they lead to the final conclusions. For example, the results on the size and number of EVs (DLS and fNTA) and the data on shuttle transcripts and shuttle proteins should be more clearly related to each other. By doing so, the reader would have a clearer understanding of the overall findings. An integrated discussion, such as "Although the size of the EVs shows no significant difference between WT and Tsc1KO mice, their content and function do differ," would be beneficial.
- While the paper does discuss the functional role of EVs, the biological significance of these findings is not fully explored. In particular, the result showing that EVs derived from Tsc1KO mice inhibit the growth of M-1 cells should be analyzed and discussed in greater detail to explain how this growth inhibition occurs.
- The conclusion of the paper summarizes the research findings. However, it should more specifically address how EVs are involved in kidney disease caused by TSC1 deficiency and provide clearer suggestions for future research. Additionally, discussing potential clinical applications and the possibility of developing new treatments targeting EVs would help emphasize the impact of the research.
Author Response
Reviewer 2
We thank the reviewer for their helpful comments and suggestions. We believe that the modifications and clarifications to the manuscript (red font) based on the reviewer’s suggestions have greatly improved this manuscript. We also hope that the revised version of this manuscript addresses the reviewer’s questions and meets their expectations.
1. This paper discusses how the genetic deletion of TSC1 and TSC2 affects the generation of exosomes (EVs). However, the background explanation is somewhat vague, and the importance and objectives of the research should be emphasized more. Specifically, a more detailed discussion of the distinct pathological effects of TSC1 and TSC2 would help readers better understand the significance of the study.
In this manuscript, we have characterized the EV derived from Tsc1KO mice. As discussed in response to Reviewer 1, we have not directly compared the shuttle content of EV from Tsc1KO and Tsc2KO mice. As indicated in the studies by Kumar et. al., 2021, the EV derived from TSC1 and TSC2 deficient cells exhibit different biological activities. Specifically, the EV from TSC2 deficient cells exhibit an mTORC1 activating capacity that is not observed in the EV from TSC1 deficient cells. As such the pro-proliferative activity associated with EV from the two sources may partially explain the increased severity of the disease caused by TSC2 vs TSC1 deficiencies. Statements/explanations regarding this comment are included in lines 265-278, 340-346, and 376-385.
Lines 46-55. The importance of this study in regard to LOH vs non-LOH mechanisms in the introduction and how EV may contribute to this is added to the description.
Lines 265-278. A discussion concerning the potential basis of increased severity of TSC2 vs. TSC1 disease.
2. The paper presents the interesting result that EVs from Tsc1KO mice exhibit anti-proliferative effects. A more detailed discussion of the mechanisms behind this, such as how specific miRNAs or proteins are involved, would make it easier to understand the role of EVs and highlight the significance of the research.
Additional discussion on the role of mRNA, lncRNA, and absence of significantly altered miRNA, as well as proteome changes, found in TSC1 EV are now included. Lines 283-288, 296-300, 306-308 and 311-313.
3. The report mentions differences in the characteristics of EVs between Tsc2KO and Tsc1KO mice. However, a more detailed analysis of this comparison is needed. It is shown that the progression of the disease in Tsc2KO mice is more severe than in Tsc1KO mice. A more thorough explanation of the reasons for this would clarify how the different functions of EVs contribute to the severity of the disease, so this should be further explored.
In this study, we compared WT vs Tsc1KO-derived EV. Direct comparison of EV from Tsc1KO vs Tsc2KO, while important, is part of our future plans. To address the reviewer’s query, we have expanded our discussion of TSC1 vs TSC2 disease severity. Direct comparison of the EV differences and their effect on the disease severity are outside the scope of this manuscript. Lines 265-278, 340-346 and 371-381.
4. The paper currently uses a variety of techniques to collect data, including Western blot, TEM, DLS, fNTA, transcriptome analysis, and proteome analysis. However, it is not entirely clear how the data from each technique are integrated and how they lead to the final conclusions. For example, the results on the size and number of EVs (DLS and fNTA) and the data on shuttle transcripts and shuttle proteins should be more clearly related to each other. By doing so, the reader would have a clearer understanding of the overall findings. An integrated discussion, such as "Although the size of the EVs shows no significant difference between WT and Tsc1KO mice, their content and function do differ," would be beneficial.
This has been discussed and added. Lines 245-247.
5. While the paper does discuss the functional role of EVs, the biological significance of these findings is not fully explored. In particular, the result showing that EVs derived from Tsc1KO mice inhibit the growth of M-1 cells should be analyzed and discussed in greater detail to explain how this growth inhibition occurs.
The section examining the potential factors in Tsc1KO derived EV and how they may affect proliferation are expanded as suggested by the reviewer. Lines 283-288, 297-300, and 311-313.
6. The conclusion of the paper summarizes the research findings. However, it should more specifically address how EVs are involved in kidney disease caused by TSC1 deficiency and provide clearer suggestions for future research. Additionally, discussing potential clinical applications and the possibility of developing new treatments targeting EVs would help emphasize the impact of the research.
Additional discussion and potential future studies to leverage our findings are now added to the concluding statement. Lines 382-392.
Round 2
Reviewer 1 Report
Comments and Suggestions for Authors
The author has answered all the questions excellently.